# Semperivium Ruthenicum Koch Extract-Loaded Bio-Adhesive Formulation: A Novel Oral Antioxidant Delivery System for Oxidative Stress Reduction

**DOI:** 10.3390/ph16081110

**Published:** 2023-08-06

**Authors:** Sebastian Mihai, Denisa Elena Dumitrescu, Antoanela Popescu, Iuliana Stoicescu, Nadia Matroud, Ana-Mădălina Răducanu, Magdalena Mititelu

**Affiliations:** 1Department of Therapeutic Chemistry, Faculty of Pharmacy, “Ovidius“ University of Constanta, 6 Căpitan Aviator Al Șerbănescu Street, 900470 Constanta, Romania; 2Department of Organic Chemistry, Faculty of Pharmacy, “Ovidius” University of Constanta, 6 Căpitan Aviator Al Șerbănescu Street, 900470 Constanta, Romania; 3Department of Pharmacognosy, Faculty of Pharmacy, “Ovidius” University of Constanta, 6 Capitan Aviator Al. Serbanescu Street, Campus, C Block, 900470 Constanta, Romania; antoanela.popescu@univ-ovidius.ro; 4Department of Chemistry and Quality Control of Drugs, Faculty of Pharmacy, “Ovidius” University of Constanta, 6 Capitan Aviator Al. Serbanescu Street, Campus, C Block, 900470 Constanta, Romania; iuliana.stoicescu@univ-ovidius.ro; 5Department of Pharmacognosy, Phytochemistry, and Phytotherapy, Faculty of Pharmacy, Carol Davila University of Medicine and Pharmacy, 6 Traian Vuia Street, 020956 Bucharest, Romania; nadia.stere@drd.umfcd.ro; 6MD Fixed Orthodontics SRL, 95 Ferdinand Blvd., 900709 Constanta, Romania; dr.raducanuanamadalina@gmail.com; 7Department of Clinical Laboratory and Food Safety, Faculty of Pharmacy, University of Medicine and Pharmacy Carol Davila, 020956 Bucharest, Romania; magdalena.mititelu@umfcd.ro

**Keywords:** bio-adhesive, periodontitis, polyphenols, antioxidant, oxidative stress, oral flora

## Abstract

Periodontitis is a chronic inflammatory disease that affects the oral cavity and can ultimately lead to tooth loss. Oxidative stress has been identified as a key factor in the development of periodontitis. In recent years, natural polyphenols have gained attention for their anti-inflammatory and antioxidant effects. This study aims to evaluate the potential of a bio-adhesive patch loaded with *Semperivium ruthenicum* Koch extract, rich in polyphenols, as a novel oral antioxidant delivery system for reducing oxidative stress in periodontitis. The plant extracts were prepared by maceration and were subjected to HPLC analysis for the identification and quantification of polyphenols. The bio-adhesive patches were prepared using a solvent-casting technique and characterized for their technical characteristics and release kinetics. The patches demonstrated satisfactory technical characteristics and followed Korsmeyer–Peppas release kinetics, with the active ingredients diffusing non-Fickian from the polymer matrix as it eroded over time. The bio-adhesive strength of the patches was comparable to other similar formulations, suggesting that the obtained patches can be tested in vivo conditions. The results suggest that treating oral periodontitis with natural polyphenols may effectively scavenge free radicals and regulate cytokine activity, leading to a reduction in oxidative stress. The non-smoking group had a mean saliva antioxidant activity of 7.86 ± 0.66% while the smoking group had a mean value of 4.53 ± 0.15%. Furthermore, treating oral oxidative stress may also contribute to overall gut health, as studies have shown a correlation between oral and gut microbiomes. Therefore, the use of bio-adhesive patches containing polyphenols may provide a promising approach for the treatment of periodontitis and its associated complications.

## 1. Introduction

The human body is colonized by a vast array of bacteria that outnumber human cells, collectively referred to as the microbiome [1]. The microbiome plays a critical role in regulating many bodily functions, including digestion, metabolism, and immune system function. Among the different parts of the body, the gastrointestinal tract hosts the most diverse population of bacteria [2]. Recent research has linked disruptions to the intestinal microbiome, known as intestinal dysbiosis, to the development of gut diseases such as Irritable Bowel Syndrome (IBS), Irritable Bowel Disease (IBD), and colorectal cancer (CRC) [3,4]. It is therefore essential to maintain a healthy balance of gut bacteria to prevent the onset of these conditions. Despite extensive research efforts, the specific microorganisms responsible for the pathogenesis of gut diseases have yet to be identified [5]. It is possible that the oral microbiome, which contains a variety of pathogenic bacteria, may serve as a reservoir for gut colonization and contribute to the development of intestinal afflictions. The gut is home to a diverse array of resident microorganisms that normally prevent colonization by exogenous organisms from external compartments [6]. However, oral pathogens have been found to colonize both the upper and lower gut, indicating that they may be involved in the pathogenesis of these afflictions [5,6].

Periodontitis is a multifactorial inflammatory disease that involves a wide range of microorganisms inhabiting the oral cavity, along with the oxidative stress caused by their metabolic processes and the host’s immune defense mechanisms [7,8,9,10,11,12]. Studies have shown that oral mechanical lesions can facilitate the spread of oral pathogens to the bloodstream [13,14]. Patients with periodontitis exhibit elevated levels of oral bacteria in their bloodstream [15], and pathogenic organisms such as *Porphyromonas gingivalis* have been identified in the blood of patients with periodontitis [16]. Certain oral bacteria implicated in periodontitis, such as *Porphyromonas gingivalis*, *Veillonella* spp., *Streptococcus* spp., and *Aggregatibacter* spp., can release H_2_S from sulfur-rich amino acids, contributing to the development of inflammation and genotoxicity. The metabolic products derived from these bacteria can also cause an increase in oxidative stress, even at low concentrations [17]. This suggests that oral pathogens may be involved in the pathogenesis of the aforementioned gut disorders [18].

Bio-adhesive delivery systems have emerged as a promising approach for the treatment of periodontitis and for reducing oral oxidative stress. The unique technical characteristics of these delivery systems offer several advantages over traditional drug formulations. Bio-adhesive delivery systems have the ability to adhere to mucosal surfaces for an extended period, leading to sustained drug release and enhanced drug efficacy [19]. This sustained release allows for a more consistent therapeutic effect and improved patient compliance, as it reduces the frequency of drug administration. Additionally, the adhesive properties of these systems allow for prolonged contact with the affected area, increasing the potential for deeper penetration and enhanced therapeutic effects.

Several studies have demonstrated the efficacy of bio-adhesive delivery systems in the treatment of periodontitis and in reducing oral oxidative stress. For instance, a recent study found that a bio-adhesive chitosan-based gel loaded with an antioxidant compound effectively reduced oxidative stress and improved periodontal parameters in a rat model of periodontitis [20]. Another study showed that a mucoadhesive patch loaded with an anti-inflammatory drug had a sustained release profile and improved the clinical outcomes of periodontitis patients [21]. The evidence from various studies supports the efficacy of bio-adhesive delivery systems in the treatment of periodontitis and in reducing oral oxidative stress, making them a promising area of research for the development of new therapeutic approaches.

This study aims to formulate and evaluate a bio-adhesive formulation loaded with an antioxidant-rich plant extract from *Sempervivum ruthenicum* Koch (a very little studied plant for therapeutic purposes) to enhance the antioxidant status of the oral cavity. Such a formulation could have significant clinical implications in the prevention and treatment of oral and gut diseases associated with microbial dysbiosis and oxidative stress.

## 2. Results

### 2.1. Total Polyphenol Content

The Folin–Ciocâlteu assay was used to determine the total polyphenol content, and the results showed that the P1 extract had an average of 7.7622 ± 0.02 mg PIR/mL, while the P2 extract had an average of 2.1939 ± 0.03 mg PIR/mL (Table 1). When compared to the dry mass of the extract obtained through solvent evaporation, the total polyphenol mass made up 78.62 ± 0.26% and 21.93 ± 0.34% of the P1 and P2 extracts, respectively.

### 2.2. ^1^H-NMR Spectroscopy

The ^1^H-NMR spectra obtained from the two types of plant extracts exhibited differences in the signal-to-noise ratio, with the P1 extract showing a better ratio. This improvement in the signal-to-noise ratio resulted in narrower signals, reducing signal overlap and enhancing the resolution for the P1 extract. The obtained spectrum exhibited characteristic signals associated with aromatic protons (δ 5.8–8.10), a methoxy group, and double bonds, consistent with the previous literature [22,23,24,25,26]. However, due to significant signal overlap, it was only possible to assign resonances to a limited number of phenolic compounds. The chemical shifts of phenolic components reported in the literature [22,23,24,25,26] were compared to our own analysis, and the results are summarized in Table 2. Despite the relatively high level of signal overlap attributed to the high concentration of compounds in the dry plant extract and the structural similarities of polyphenols, Figure 1 illustrates distinct peaks in the aromatic regions as well as in the alcohol and aldehyde regions of the spectrum.

### 2.3. HPLC Assay

The HPLC assay conducted on the plant extract revealed a significant amount of bioactive compounds, including polyphenols, flavonols, heterosides, and flavones, as illustrated in Figure 2 and Table 3. Among the polyphenolic acids, Gallic acid displayed the highest concentration in both types of plant extracts, followed by Ellagic acid. While the flavonols quercetin, kaempferol, and isorhamnetin were also detected, their concentrations were relatively low. In contrast, Astragalin and Rutin were present in relatively high concentrations and were successfully quantified. It is worth noting, however, that the extract from fresh plant material had significantly lower concentrations of bioactive compounds compared to the dry plant extract. These results are consistent with previously published works [27,28] and indicate that the *Sempervivum ruthenicum* Koch dry plant extract is a rich source of various bioactive compounds, especially polyphenols and flavonoids, which may offer health benefits upon consumption.

### 2.4. Xanthine Oxidase and Albumin Denaturation Inhibitory Activity

The xanthine oxidase inhibitory activity demonstrated notable outcomes, as presented in Table 4. The extract derived from the freshly harvested plant material exhibited a more pronounced inhibitory effect on the enzyme, indicating the degradation of the active constituents responsible for this activity during the drying process, likely due to oxidation. The inhibitory effect displayed a dose-dependent and linear relationship, as depicted in Figure 3, suggesting a competitive mechanism of action for xanthine oxidase inhibition.

The albumin denaturation inhibitory activity exhibited notable results, surpassing those demonstrated by the reference standard. The data presented in Table 5 emphasize the higher inhibitory activity of the extract obtained from the dried plant material. Interestingly, both extracts displayed similar IC_50_ values, indicating that the active compounds responsible for this activity are present in comparable concentrations in both types of extracts.

### 2.5. Bio-Adhesive Patch Evaluation

#### 2.5.1. Mass Uniformity, Patch Thickness, and Flexibility

After examining the weight parameters, it was found that the patches had a consistent weight, with an average of 4.28 ± 0.148 mg/cm^2^ for the P1 formulation and 4.27 ± 0.116 mg/cm^2^ for the P2 formulation. The thickness of the patches was also consistent, with an average thickness of 0.0087 ± 0.004 mm for the P1 formulation and 0.088 ± 0.004 mm for the P2 formulation. Additionally, all patches tested showed a breaking resistance of over 200 folds, which meets the quality standards outlined in the European Pharmacopoeia 9th Edition [29].

#### 2.5.2. Surface pH and Swelling Index

After hydrating the oral patches the surface pH was determined, as well as the swelling index. Both types of formulations had a 5.5 surface pH. The swelling index was 313.635 ± 15.65% and 329.775 ± 8.83% for the P1 and P2 formulations, respectively. These values show a high capacity for holding water in the patch matrix, which can facilitate the diffusion of the active principles from the polymer matrix and into the adjacent mucosa. It is also worth mentioning that all formulation retained their integrity throughout the assay procedures.

#### 2.5.3. Uniformity of Content

After conducting a spectrophotometric assay at 290 nm and 215 nm, for the P1 and P2 formulations, respectively, the extract content for the P1 formulation was determined to be 6.9592 ± 0.004 mg, and for the P2 formulation, it was 7.0117 ± 0.003 mg. However, upon assessing the total polyphenols using the Folin–Ciocîlteu method, the total polyphenol content of the formulations was quantified as 1.467 ± 0.002 mg Pir/10 cm^2^ oral patch for the P2 formulation and 5.2443 ± 0.004 mg Pir/10 cm^2^ oral patch for the P1 formulation, respectively. Based on these results, and on the calculated total phenol content of the dry plant extracts, the loading efficiency for the total polyphenols was calculated to be 95.38 ± 0.24% for the P1 formulation and 95.75 ± 0.33% for the P2 formulation, respectively.

The HPLC assay revealed the presence of several polyphenols, including gallic acid, caffeic acid, ferulic acid, astragalin, kaempferol, quercetin, and rutin. After integration, a total of 1.124 mg polyphenols were quantified from the P2 formulation, and 1.1422 mg from the P1 formulation, respectively. A representative chromatogram and detailed quantification of the identified polyphenols can be seen in Figure 4 and Table 6.

#### 2.5.4. Ex-Vivo–Bio-Adhesive Strength

The adhesive strength of the patches is a crucial factor that indicates their effectiveness and quality. It ensures the extended contact time necessary for the sustained release of active ingredients. The results showed that both formulations had a bio-adhesive strength of 0.0749 ± 0.0022 N, and no significant differences were observed between the two formulations.

#### 2.5.5. In Vitro Release Study

The Korsmeyer–Peppas model is commonly used to study the release kinetics of drugs from polymer matrices. The model describes drug release from a polymer matrix as a function of time and can be expressed as follows:M_t_/M_inf_ = k × t^n^(1)
where M_t_ is the amount of drug released at time t, M_inf_ is the total amount of drug in the matrix, k is a constant that depends on the characteristics of the matrix and drug, and n is the release exponent.

The release exponent, n, is used to determine the mechanism of drug release from the polymer matrix. A value of n = 0.5 indicates that the drug release is governed by diffusion, while a value of n > 1 indicates a Super Case II, which indicates an extreme form of transport where the outer layer of gel limits the vitreous nucleus, which prevents the axial swelling of the gel and produces tension of compression on the nucleus. As the polymeric interface moves to the nucleus, the increasing tension will lead to the nucleus breaking. The value of n can be used to predict the drug release mechanism and optimize the drug release rate and duration [30].

In this study, both formulations showed non-Fickian (anomalous) diffusion of the active principles from the polymer matrix, as both values of n were >1 (Figure 5). This suggests that the release mechanism is not solely dependent on the diffusion of the active principles through the polymer matrix, but also on other factors such as swelling, erosion, and relaxation of the polymer matrix, and corresponds to a zero-order kinetics release type. When n > 1, the release rate decreases more slowly over time, indicating a slower rate of drug diffusion from the matrix. This suggests that the drug is bound more tightly to the polymer, and the release mechanism is influenced by factors such as chain relaxation or swelling of the polymer.

Overall, the Korsmeyer–Peppas model with n > 1 provides a more accurate representation of the release kinetics of drugs or other substances from polymeric matrices and can help optimize drug delivery systems for specific applications [31].

#### 2.5.6. In Vivo Oral Antioxidant Effect

The study group consisted of 48 healthy volunteers aged between 20 and 35 years, with an average age of 27.41 years. After evaluating the oral residence time of the oral patches, there were no significant differences between the three sub-groups, suggesting that the plant extracts do not impact adhesion. The maximum oral residence time was 274 min and the minimum recorded was 110 min, suggesting a high variability between applications, which can be attributed to each subject’s oral activity (tongue movements, amount of saliva produced, etc.). The mean time of residence was 176.875 ± 41.97 min.

After evaluating the saliva activity against the DPPH free radical, several differences could be recorded between subgroups before applying the patches and after. The non-smoking group had a mean saliva antioxidant activity of 7.86 ± 0.66% while the non-smoking group had a mean value of 4.53 ± 0.15%.

Upon application of the patches, a decline in antioxidant activity was initially observed across all subgroups, followed by a consistent increase that peaked at 25.7% inhibitory activity. The P1 subgroup exhibited greater antioxidant activity in saliva, and the smoking subgroup demonstrated a higher increase in antioxidant activity compared to non-smokers. Comprehensive findings are presented in Table 7 and Figure 6.

Significant positive correlations were established (r > 0.6500) between the in vitro release profile of the patches and the antioxidant activity, as seen in Appendix A, Figure A1, Figure A2 and Figure A3, which suggests an optimal formulation and release profile for attaining the antioxidant effect.

## 3. Discussion

Periodontitis is a multifactor inflammatory disease that affects the support structures of the teeth and ultimately leads to the erosion of the alveolar bone and teeth loss [7,8]. The microbes that colonize the sub-gingival dental plaque play an essential part in the pathogenesis of the disease, along with the host inflammatory response, which, in part, reacts with the proteolytic and lipopolysaccharide enzymes produced by the resident flora [32,33].

Decades ago, it was suggested that oxidative stress plays a role in the development of periodontitis [10]. However, it was not until recently that oxidative stress and reactive oxygen species (ROS) garnered attention as crucial factors in various inflammatory diseases.

The extract derived from the dried plant material demonstrated a higher concentration of polyphenols, which can be attributed to the increased solvent penetration of the dry plant material. The quantification of phenolic compounds was performed using HPLC-DAD, employing a standard method, and the results were further confirmed by 1H-NMR analysis, which revealed characteristic peaks in the spectrum. The existing literature [23,24,25,26] supports the presence of polyphenol peaks in the regions described in this study, indicating the presence of aromatic rings and hydroxyl groups in the complex mixture. This observation aligns with the findings of the current study, further validating the identification and characterization of polyphenolic compounds in the extract. However, due to peak overlapping, only 10 compounds could be identified, out of which 8 were both confirmed and quantified by HPLC. Additionally, a Folin-Ciocîlteu assay indicated high concentrations of polyphenols in the extract obtained from the dry plant material. Notably, the P1 extract exhibited higher polyphenol concentrations after HPLC-DAD quantification, suggesting a more efficient recovery of phenolic compounds from the dried plant material.

The patches obtained exhibited a smooth and uniform surface, devoid of any noticeable particle agglomerations or discoloration. Upon completion of the assay, both formulations demonstrated satisfactory technical characteristics, indicating an optimal workflow for their preparation. The release of the active ingredients followed a Korsmeyer–Peppas release kinetics, with the active ingredients diffusing in a non-Fickian manner (Super Case II) from the polymer matrix as it eroded over time. This type of release kinetics is typical for bio-adhesive formulations used on the mucosa [31]. In this scenario, the release of the substance is not solely governed by diffusion, but rather a combination of mechanisms, such as swelling, erosion, or relaxation of the polymeric matrix. As a result, the release kinetics are complex and cannot be described by simple mathematical models.

The assessed bio-adhesive strength of the formulations is comparable to other similar formulations [31,32,33,34], suggesting that the obtained patches can be tested in vivo conditions, without the risk of losing adhesion or disintegration before the complete release of the active ingredients.

The formulations had a total polyphenol loading efficiency of >95%; however, the HPLC assay was not able to identify and quantify all the compounds previously identified in the dry plant extracts. This may be due to the presence of polymers, or it may be that some of the compounds reacted or oxidized during the preparation of the bio-adhesive films. The stability of the bioactive compounds in the formulations warrants special interest in the future.

In recent times, natural polyphenols have demonstrated noteworthy progress in their anti-inflammatory and antioxidant effects. These compounds exhibit various mechanisms that effectively scavenge free radicals and regulate cytokine activity. Furthermore, polyphenols directly prevent the reduction of Fe^3+^, which results in a significant reduction in the free-radical concentration. In addition to their antioxidant properties, polyphenols also exhibit a protective effect on inflammation by modulating the NLRP3 inflammasome [35]. A recent study by Hori et al. [36] revealed that green propolis, which is abundant in cinnamic acids, effectively suppresses the secretion of IL-1β mediated by inflammasomes, along with the activation of caspase-1. Additionally, polyphenols have demonstrated their ability to reduce inflammation by acting as a regulator, attenuating the activation effect of pro-inflammatory cytokines such as NF-κβ.

Inhibition of xanthine oxidase can lead to a reduction in the production of superoxide radicals, as supported by previous research [37]. In this study, both plant extracts exhibited high inhibitory activity against xanthine oxidase. However, it was observed that the drying process potentially influenced this activity, possibly through the oxidation of the responsible compounds.

The denaturation of albumin can contribute to Type III hypersensitivity reactions, which play a role in chronic inflammation [38]. Additionally, it has been demonstrated that non-steroidal anti-inflammatory drugs not only inhibit cyclooxygenase but also prevent protein denaturation [38]. The results of our study demonstrated excellent albumin denaturation inhibitory activity, surpassing that of the reference standard. It is important to note that similar findings regarding the albumin denaturation inhibitory activity of plant extracts have been reported by other authors [38,39,40,41,42,43,44].

The oxidative stress in the oral cavity is caused by both the metabolic activity of the endogenous flora and certain exogenous factors such as smoking. However, a recent study observed a higher level of antioxidant activity in the saliva of the smoking group compared to the non-smoking group. It is worth noting that several previous studies [45,46] have consistently reported lower antioxidant levels in the saliva of smokers compared to non-smokers. The discrepancy in the findings of this study may be attributed to the relatively young age of the participants and the possibility of compensation mechanisms in the host antioxidant defense.

Oxidative stress has been identified as a crucial factor in the development of periodontitis, and the antioxidant properties of polyphenols, combined with a prolonged contact time with the mucosa, have been shown to effectively scavenge free radicals and regulate cytokine activity. Furthermore, the use of bio-adhesive patches containing polyphenols has shown promising results in in vitro release and adhesion strength, suggesting the possibility of in vivo applications. It is worth noting that treating oral oxidative stress may also contribute to overall gut health, as studies have shown a correlation between oral and gut microbiome [47,48].

## 4. Materials and Methods

### 4.1. Methods

#### 4.1.1. Plant Extracts

The plant material, *Sempervivum ruthenicum* Koch, was harvested from the Dobrogea region of Romania (44°30′17.5″ N 28°25′41.4″ E) while the plant went through its flowering stage (Figure 7). The plant material was macerated fresh and after drying in standard conditions (loss upon drying 87.75%). A voucher specimen was deposited at the Botany Chair of the Faculty of Pharmacy, “Ovidius” University, Constanta, Romania.

The *Sempervivum ruthenicum* Koch extracts were obtained via maceration for a period of 14 days with daily agitation in a water–ethanol mixture (50/50 *v*/*v*). The raw extracts were then filtered through Whatman paper (no. 14) to obtain clear solutions. Two types of extracts were prepared, one from dry plant material (P1) and the other from fresh material (P2), taking into account the loss of 87.75% upon drying. The concentration of both extracts was 100 mg of plant product per milliliter. The clear extracts were vacuum-dried in an IKA RV10 rotary evaporator (Staufen, Germany) at a temperature of 50 °C. The resulting dry extracts were ground into a fine powder and stored in a desiccator for future use.

#### 4.1.2. Polymers and Reagents

The oral bio-adhesive patches were formulated using polymers such as gelatin (strength 300, type A), pectin derived from apple, polyvinylpyrrolidone, and methylcellulose. These polymers, as well as all other reagents used in the formulation process, were obtained from Sigma-Aldrich and were of analytical grade. All other reagents used were of analytical grade and were purchased from Sigma-Aldrich, Munich, Germany.

### 4.2. Total Polyphenols Content Analysis

The total polyphenol content of the dried extract was determined using the modified Folin–Ciocâlteu method. To prepare the stock solutions, 100 mg of each extract was dissolved in 10 mL of double-distilled water. Subsequently, 100 μL of the stock solution was mixed with 500 μL of distilled water and 100 μL of Folin–Ciocâlteu Reagent. The mixture was left standing for 6 min, then 1000 μL of 5% CaCO_3_ was added, followed by 500 μL of distilled water. The mixture was vortexed and left to settle for 90 min, after which the absorbance was measured at 760 nm using a Varian Cary 50 UV-VIS spectrophotometer (Palo Alto, CA, USA).

The standard curve was drawn using pyrogallol in 6 gradually different dilutions (R^2^ > 0.95). The total phenolic content was calculated as pyrogallol equivalents, and all assays were performed in triplicate.

#### Plant Extracts Maximum Absorbance

To determine the maximum absorbance of the plant extract, UV-VIS spectrometry was conducted using a Varian Cary 50 spectrophotometer with Varian Scan Kinetics software version 6.2. The stock solution prepared from the dried extracts was diluted ten times, and the maximum absorbance of the extract was recorded for each dilution. The spectral data were compiled and subtracted to obtain the final maximum absorbance wavelength. After scanning the UV-VIS full spectrum, the optimal absorbance for the P1 and P2 extracts was found to be 290 nm and 215 nm, respectively. The calibration curves were drawn according to the data presented in Table 8 and Figure 8.

A test run was performed using individual polymers at these specific wavelengths, and no high absorbance was recorded for any other formulation component.

### 4.3. ^1^H−NMR Spectroscopy

The spectra were acquired using a Varian Mercury 300 spectrometer (Agilent Technologies, Boblingen, Germany) employing a standard s2 pulse experiment with water suppression. Prior to analysis, the dry samples were dissolved in a mixture of deuterated dimethyl sulfoxide (DMSO) and trifluoroacetic acid at a volume ratio of 3:1 (*v*:*v*). The acquisition parameters were set as follows: The scans were collected to obtain data points across a spectral width of 10,000 Hz, with a relaxation delay of 5 s and an acquisition time of 2.7 s. To ensure accurate interpretation, the spectra were subjected to phase correction using Mnova 14.3.3 software (Mestrelab Research, Santiago De Compostela, Galicia, Spain). Additionally, manual baseline correction was performed utilizing the same software to enhance data quality and remove any systematic distortions.

### 4.4. HPLC Assay

In order to determine the bioactive compound present in the plant extract, a standardized HPLC method was utilized, which has been described in the USP 30-NF25 Pharmacopoeia [49]. The HPLC system used consisted of an Agilent 1200 chromatogram equipped with a quaternary pump, DAD, thermostat, degas system, and autosampler (Agilent Tehnologies, Boblingen. Germany). A C18 Zorbax XDB column (250 mm × 4.6 mm; 5 µm) was employed, and the eluents were composed of 0.1% phosphoric acid (A) and acetonitrile (B), with a linear gradient of 10% B for 13 min, 22% B for 1 min, 40% B for 3 min, and 10% B for 1 min. The flow rate was set to 1.5 mL/min and the column temperature was maintained at 35 °C. The DAD system was set to detect wavelengths of 310 nm, 335 nm, and 360 nm simultaneously. The standards used in the study were purchased from Chromadex (Wesel, Germany) and included E-resveratrol, Z-resveratrol, caffeic acid, chlorogenic acid, cinnamic acid, ellagic acid, vanillin, gallic acid, ferulic acid, astragalin, isorhamnetin, kaempferol, scutellarin, rutoside, and quercetin. All assays were run in triplicate, and the results were expressed as mean ± SD.

### 4.5. Xanthine Oxidase Inhibitory Activity

The inhibitory effect of xanthine oxidase was measured using spectrophotometry at a wavelength of 290 nm under aerobic conditions, following the method described by Isa and Mohamad with slight modifications [50]. The experimental setup consisted of a reaction mixture containing 1 mL of plant extract at concentrations of 100, 80, 60, 50, 40, and 20 μg/mL, 2.9 mL of phosphate buffer (pH 7.5), and 0.1 mL of freshly prepared xanthine oxidase solution (Sigma-Aldrich, Munich, Germany) (0.01 U/mL in phosphate buffer).

To initiate the reaction, the samples were pre-incubated at 25 °C for 15 min, followed by the addition of 2 mL of a freshly prepared 150 mM xanthine solution (Sigma-Aldrich, Munich, Germany) in phosphate buffer (pH 7.5). The reaction mixture was then incubated at 35 °C for 30 min. The reaction was quenched by adding 1 mL of a 1 M HCl acid solution. The absorbance of the reaction mixture was measured at 290 nm using a Varian Cary 50 spectrophotometer (Agilent Technologies, Boblingen, Germany).

Blank samples were prepared as follows: B1 by replacing xanthine oxidase with phosphate buffer, B2 by adding the enzyme solution after the addition of HCl, and B3 by substituting the test sample and enzyme solution with an equal amount of phosphate buffer. A positive control, allopurinol (Sigma-Aldrich, Munich, Germany), was used with the same concentrations as the samples.

The inhibition percentage was calculated using the formula proposed by Umamaheswari et al. [51]:% inhibition = [(A − B) − C − D)]/A − B(2)
where A represents the absorbance of B2, B is the absorbance of B3, C is the absorbance of the test samples, and D is the absorbance of B1. This formula was employed due to the high absorbance exhibited by the plant extracts at 290 nm, which interfered with the assay. By using this formula, the interferences caused by the extracts were minimized.

All experiments were conducted in triplicate, and the results were expressed as the median ± standard deviation. The IC50 values, representing the concentration of the extract required to inhibit 50% of xanthine oxidase activity, were calculated by plotting the extract concentrations against the inhibitory percentage.

### 4.6. Albumin Denaturation Inhibitory Activity

The inhibitory activity against albumin denaturation was determined using the method proposed by Khan et al. [52]. In brief, freshly prepared albumin from Sigma-Aldrich (Munich, Germany) in phosphate-buffered saline (pH 6.4) was mixed with 2 mL of plant extracts at concentrations of 125, 250, 500, 750, and 1000 μg/mL, along with 2.8 mL of phosphate-buffered saline (pH 6.4). The reaction mixtures were vortexed and incubated at 37 °C for 15 min, followed by heating at 70 °C for 5 min. Subsequently, the solutions were cooled to room temperature, and the absorbance was measured at 660 nm using a Varian Cary 50 UV-Vis spectrophotometer (Agilent Technologies, Boblingen, Germany). A control sample was prepared by replacing the plant extract with double-distilled water. As a positive control, acetylsalicylic acid (Sigma-Aldrich, Munich, Germany) was used. The percentage of inhibition was calculated using the following formula:%inhibition = [(A_control_ − A_sample_)/A_control_] × 100(3)
where A_control_ is the absorbance of the control sample, and A_sample_ is the absorbance of the test sample.

### 4.7. Preparing the Bio-Adhesive Patches

Bio-adhesive oral patches loaded with dried plant extracts were prepared using a modified method proposed by Hashemi et al. [53]. In brief, gelatin was dissolved in 15 mL of double-distilled water at 60 °C with constant stirring. Then, polyvinylpyrrolidone and pectin were added and stirred until a clear solution was formed. Propylene glycol was added to the clear solution, and stirring was continued while heating was stopped. In a separate beaker, methylcellulose was dispersed in 15 mL of double-distilled water under magnetic stirring until a clear, viscous gel formed. The two solutions were mixed under magnetic stirring at 150 rpm and 22 °C for three hours to ensure complete homogenization. The dried plant extracts were dissolved in 3 mL of a water–ethanol mixture (50/50 *v*/*v*) and added to the polymer solution under magnetic stirring. The resulting mixtures were stirred at room temperature for two hours. Then, the mixtures were poured into glass Petri dishes (9.1 cm diameter) and dried at 40 °C for 15 h. The resulting patches were carefully detached from the Petri dishes, covered in aluminum foil, and stored in a desiccator until further use. The quantities used for the preparation of the bio-adhesive patches are presented in Table 9.

Following the evaporation of the solvent, patches with a collective surface area of 85,795 cm^2^ were acquired. The dry plant extract concentration was determined to be 0.6993 mg/cm^2^ for both formulations. The patches exhibited a smooth surface with no observable conglomerates. The P1 formulation displayed a brownish coloration, while the P2 formulation had a faint yellowish hue (Figure 9).

### 4.8. Bio-Adhesive Patch Evaluation

To evaluate the oral patches, 1 cm^2^ sections were excised using a metal punch. Mass uniformity was assessed in accordance with the Romanian Pharmacopoeia 10th Edition [29] by precisely weighing 10 patches with a 1 cm^2^ area on a Mettler Toledo AG3204 precision scale for each formulation and determining the mean and standard deviation. The thickness of the patches was measured by selecting 10 patches at random from both formulations and using a Torpex micrometer. Patch flexibility was determined by folding three patches repeatedly until they broke.

#### 4.8.1. Patch Surface pH

The surface pH of the polymer patches was measured utilizing a modified version of the method described by Bottenberg et al. [54]. Initially, a 2% (m/V) agar solution was prepared by dispersing the agar in a freshly prepared phosphate buffer (pH 6.8), and the mixture was then poured into a Petri dish and allowed to solidify. Next, the polymer patches were placed on the surface of the agar and allowed to hydrate for two hours. Finally, the pH was measured by touching the surface of the patches with the glass tip of a digital pH meter (Elico LI 120, Hyderabad, India).

#### 4.8.2. Uniformity of Content

The uniformity of content was determined by placing a 10 cm^2^ patch in a beaker containing 100 mL of double-distilled water and stirring the mixture at 37 °C until the patch was fully dissolved. From the resulting solution, 1 mL samples were collected, and at 215 nm PVP, the absorbance was measured at the maximum absorption wavelength (290 nm for PVU nm) of the extract. Calibration curves that had been previously established were used to calculate the concentration of each type of extract. Subsequently, samples were drawn from the solution and assayed using the Folin–Ciocîlteu and HPLC methods described above.

#### 4.8.3. Bio-Adhesive Patch Swelling Index

To determine the swelling index of the bio-adhesive patches, ten samples were placed on the surface of a Petri dish containing a 2% agar solution. The dishes were then incubated at 37 °C, and the samples were weighed after 120 min of incubation. The swelling index was subsequently calculated using the following formula:swelling index (%) = [(M_t_ − M_0_)/M_0_] × 100,(4)
where M_0_ represents the initial mass of the patch and M_t_ represents the mass of the inflated patch after incubation for t (time).

#### 4.8.4. Determining the Bio-Adhesive Strength

To determine the bio-adhesive strength of the patch, a physical scale adapted from Gupta’s specifications [55] was used. The patch was first fixed onto a flat glass surface with double-sided tape, which was then attached to the apparatus scale. A piece of freshly harvested lamb oral mucosa was washed with phosphate buffer (pH 6.8) and placed directly beneath the fixed patch. The apparatus was set up so that the patch would be in full contact with the oral mucosa for 30 s, with a 5 g weight pressing down on it. A pre-weighed Berzelius beaker was placed on the second scale and gradually filled with water until the patch detached from the oral mucosa. The added water was then weighed and added to the beaker mass to obtain the final mass of detachment (M). The bio-adhesive strength was subsequently calculated using the formula:Bio-adhesive strength (N/m) = (M × g)/1000,(5)
where M is the mass required to detach the oral patch from the mucosa and g represents the gravitational acceleration.

#### 4.8.5. In Vitro Release Study

The in vitro release of the bio-adhesive patch formulations was determined using a USP type-2 paddle apparatus (Vankel VK 7000, Erweka, Germany). The dissolution medium comprised 500 mL of phosphate buffer (pH 6.2) maintained at a temperature of 37 ± 1 °C with continuous stirring at 50 rpm. A 10 cm^2^ sample from each formulation was placed on the surface of the dissolution medium. At predetermined time intervals (5, 10, 15, 30, 60, 90, and 120 min), 2 mL samples were drawn, and the volume was replenished with fresh buffer solution. The drawn samples were analyzed spectrophotometrically at the previously determined maximum absorption wavelength. The release kinetics were determined using modeling of the release pattern with the most suitable R^2^.

#### 4.8.6. Bio-Adhesive Patch In Vivo Antioxidant Activity

The study recruited 48 healthy volunteers between the ages of 20 and 35 who gave their informed consent in accordance with the WHO Helsinki Declaration (revised in Edinburgh 2000) and with approval from the Bioethics Committee of the University of “Ovidius” Constanta, Romania (approval no. 17712/12.11.2018). Volunteers had to meet certain inclusion criteria such as being clinically healthy and having no allergies to the polymers in the formulations or to the plant species being tested. Alkaloid tests were performed on the plant extracts with negative results, and no toxic metabolites were found in the literature review of the *Sempervivum genus*.

The volunteers were divided into two groups based on their smoking status, and each group was further divided into three sub-groups: One receiving patches without plant extract (Control), one with patches containing dry plant extract (P1), and one with patches containing fresh plant extract (P2). Volunteers were instructed to abstain from brushing their teeth and using certain products on the day of the experiment. The smoking group was also instructed not to use tobacco-derived products on the day of the experiment. All volunteers received 350 mL of water on an hourly basis throughout the experiment.

Bio-adhesive patches with an area of 1 cm^2^ were applied to the buccal mucosa of all volunteers for 20 s. Saliva samples were collected at specific time intervals and immediately centrifuged for 10 min at 13,000 rpm. The supernatant was used for antioxidant assays. The DPPH inhibitory activity (%) was calculated using the following formula:[(A_blank_ − A_sample_)/A_sample_) × 100],(6)
where A_blank_ is the absorbance of a DPPH solution mixed with 100 μL of the phosphate buffer (pH 6.8) and A_sample_ is the absorbance of the sample at 517 nm. Variance analysis and Student’s T-test were performed for statistical analysis of the data, with a significance level of *p* < 0.05. This allowed for the evaluation of any differences between the Control, P1, and P2 groups. The statistical analysis helped to determine the effectiveness of the bio-adhesive patches containing the plant extract in raising the antioxidant status of the oral cavity.

All reagents used in analyses were obtained from Sigma-Aldrich and were of analytical grade.

## 5. Conclusions

The findings of this study support the potential of natural polyphenols formulated in an oral bio-adhesive patch for the treatment of periodontitis, a disease that can ultimately lead to tooth loss. Hence, there is a need for additional research to explore the potential of bio-adhesive formulations incorporating polyphenols and other bioactive compounds in the treatment of periodontitis and other inflammatory diseases. Such investigations could have significant implications for human health beyond the oral cavity, as systemic inflammation has been linked to a range of chronic conditions, including cardiovascular disease, diabetes, and cancer. Therefore, exploring the potential of bio-adhesive delivery systems for delivering polyphenols and other bioactive compounds could pave the way for the development of novel therapeutic strategies for addressing a broad range of health issues. The experimental results highlight the therapeutic potential of the plant *Sempervivum ruthenicum* Koch, which, until now, has been very little studied.

## Figures and Tables

**Figure 1 pharmaceuticals-16-01110-f001:**
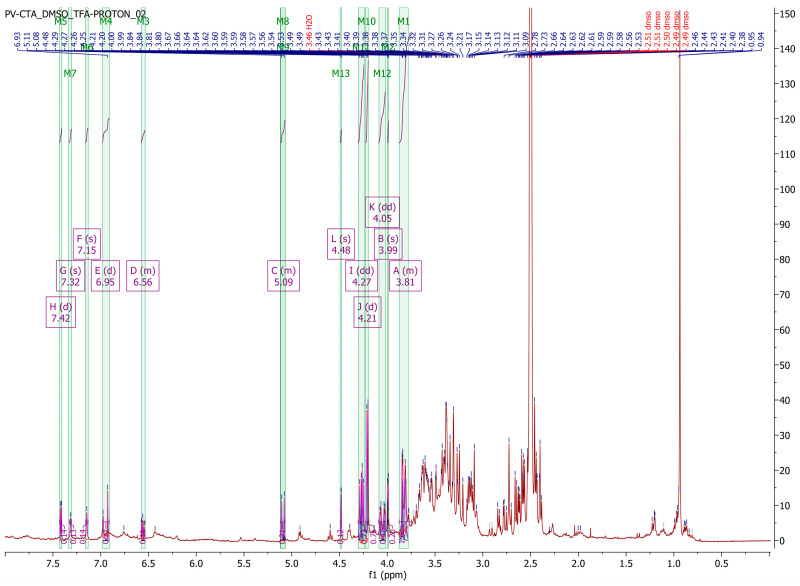
(**Top**) ^1^H-NMR spectrum of the P1 plant extract. (**Bottom**-**Left**) Enhanced view of the aromatic region. (**Bottom**-**Right**) Enhanced view of the alcohols-aldehydes region.

**Figure 2 pharmaceuticals-16-01110-f002:**
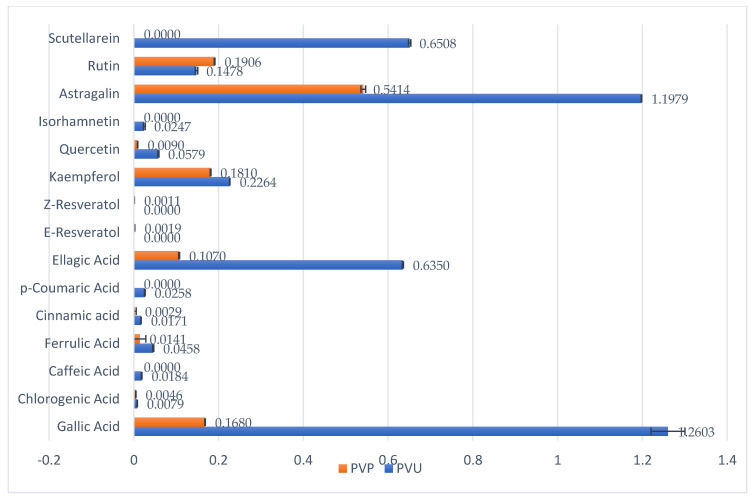
*Sempervivum ruthenicum* Koch phenolic compounds.

**Figure 3 pharmaceuticals-16-01110-f003:**
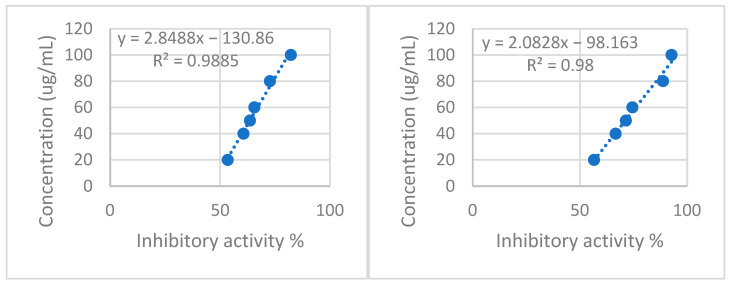
(**Left**) P1 xanthine oxidase inhibitory activity. (**Right**) P2 Xanthine oxidase inhibitory activity.

**Figure 4 pharmaceuticals-16-01110-f004:**
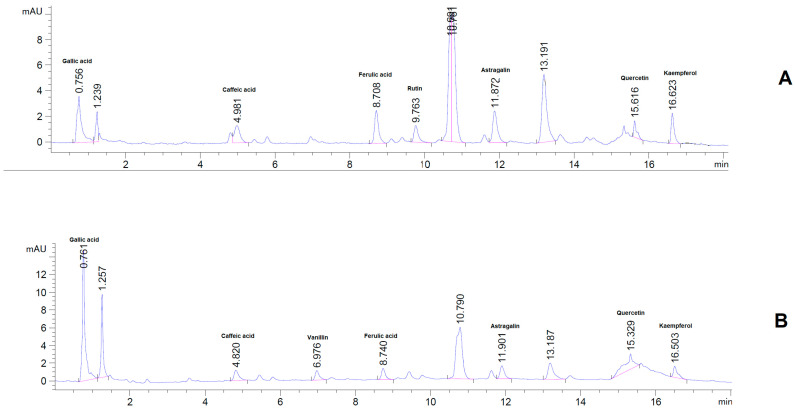
Representative HPLC chromatogram for the oral bioadhesive formulations. (**A**). P1; (**B**). P2.

**Figure 5 pharmaceuticals-16-01110-f005:**
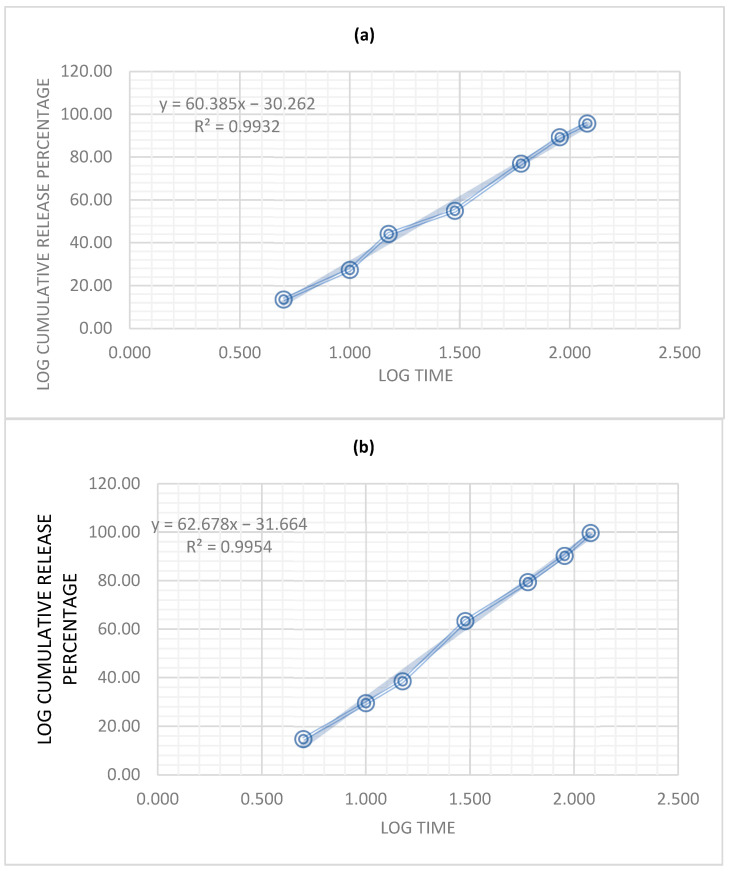
Korsmeyer–Peppas plot for: (**a**) P1, (**b**) P2.

**Figure 6 pharmaceuticals-16-01110-f006:**
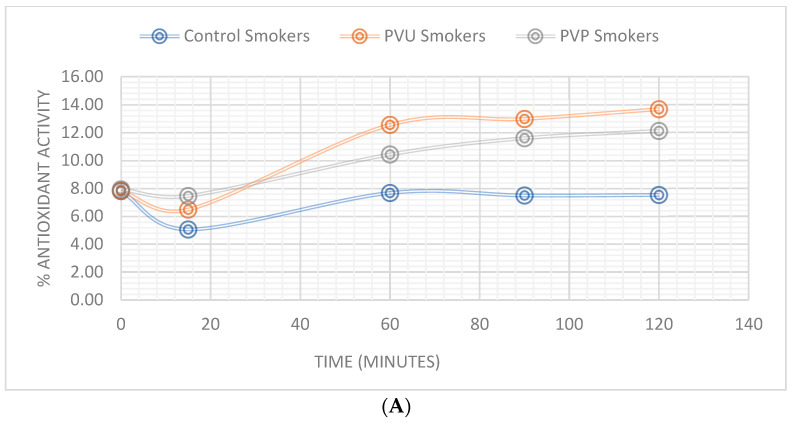
Percentage of total saliva antioxidant activity: (**A**) Smokers, (**B**) non-smokers.

**Figure 7 pharmaceuticals-16-01110-f007:**
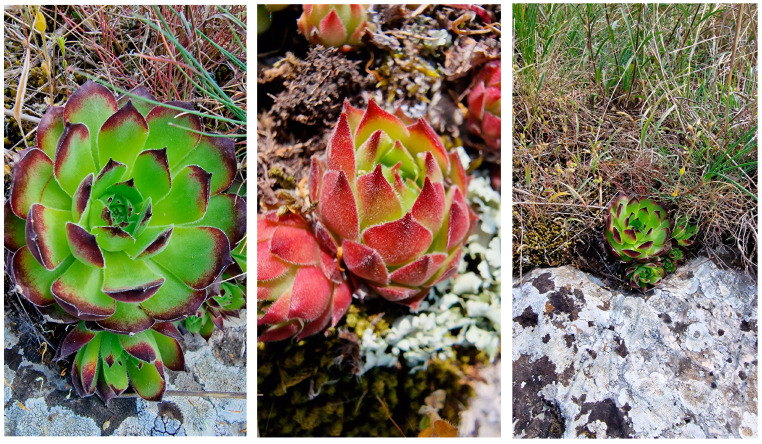
*Sempervivum ruthenicum* Koch.

**Figure 8 pharmaceuticals-16-01110-f008:**
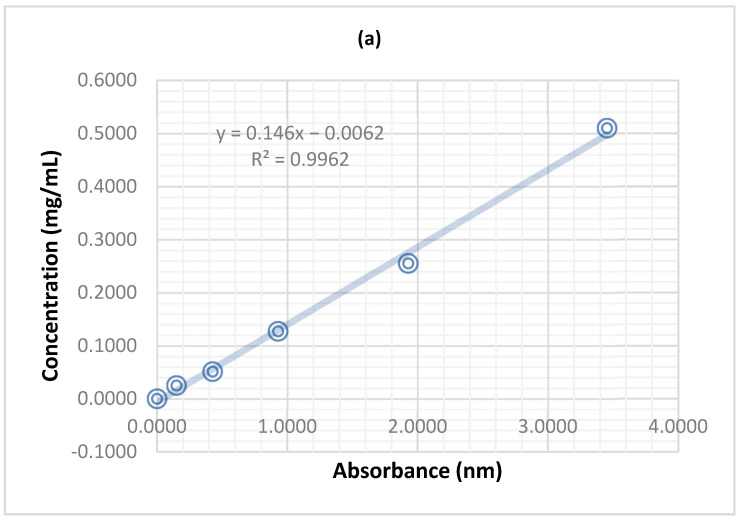
The calibration curves for the two types of extracts: (**a**) P1 at 290 nm (**b**) P2 at 215 nm.

**Figure 9 pharmaceuticals-16-01110-f009:**
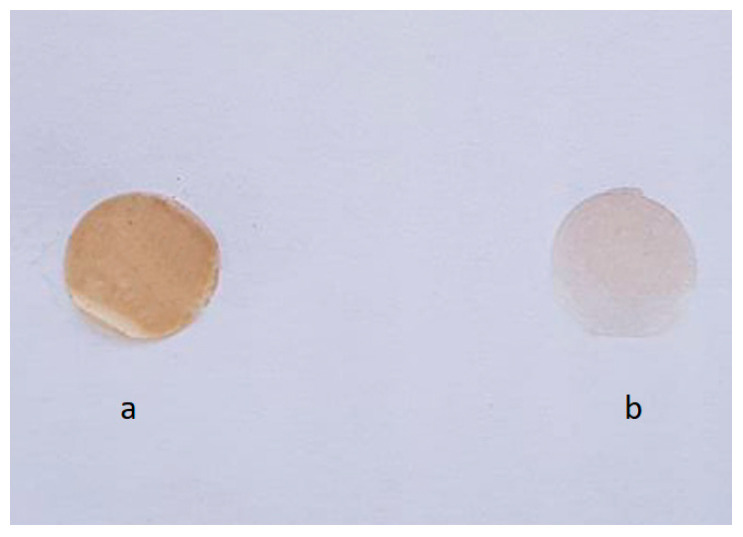
Obtained oral patches with a 1 cm^2^ surface area: (**a**) P1, (**b**) P2.

**Table 1 pharmaceuticals-16-01110-t001:** The total polyphenol content of the two dry plant extracts.

Run No.	Polyphenol Content (mg PIR/mL)		Polyphenol Percentage to Dry Extract Weight	
	P1	P2	P1	P2
1	7.8330	2.2324	78.33	22.32
2	7.8691	2.1826	78.69	21.82
3	7.8844	2.1667	78.84	21.66
Average	7.8622	2.1939	78.62	21.93
Standard deviation	0.026	0.03	0.26	0.34

**Table 2 pharmaceuticals-16-01110-t002:** The chemical shifts of phenolic components reported in the literature [22,23,24,25,26] in comparison to our own analysis.

Compound	Simulated Spectrum [22] *	Anastasiadi et al. [23]	Lopez-Martinez et al. [24]	Savage et al. [25]	Kim et al. [26]	Our Results
Gallic acid	δ 7.24 (2H, d)	1H NMR: δ 7.08 (s, H2, H6)	1H NMR: δ 6.91 (H2, H6)	7.15 (s)	NA	δ 7.15 (s, H2, H6)
Syringic acid	δ 3.80 (6H, s), 7.26 (2H, d)	δ 7.32 (s, H2, H6), 3.87 (s, OMe3, OMe5)	NA	NA	NA	δ 3.99 (s, OMe), 7.32 (s, H2, H6)
Tartaric acid	1H NMR: δ 4.94 (2H, d).	NA	NA	4.36 (s)	NA	4.48 (s)
Chlorogenic acid	δ 1.86 (1H, dd), 2.09 (1H, dd), 2.19–2.46 (2H, 2.26 (dd), 2.39 (dd)), 3.22–3.42 (2H, 3.27 (q,), 3.35 (dd)), 4.85 (1H, td), 6.50 (1H, d), 6.77 (1H, dd), 7.28 (1H, dd), 7.63–7.81 (2H, 7.69 (dd), 7.74 (d)).	NA	δ 6.99 (H2, H6), 7.42 (H-alfa), 3.91 (H1)	δ 3.88 (m), 6.96 (d), 7.15 (dd),	NA	δ 3.81 (m, H1), 6.56 (m, H-beta), 6.95 (d, H2, H6), 7.42 (d, H-alfa)
Caffeic acid	δ 6.45 (1H, d), 6.77 (1H, dd), 7.28 (1H, dd), 7.62–7.79 (2H, 7.68 (dd), 7.72 (d)).	δ 7.53 (d, H-alfa), 6.93 (dd, 6),	δ 6.96 (H2, H6), 7.41 (H-alfa)	NA	NA	δ 6.95 (d, H2, H6) 7.42 (d, H-alfa)
Ferulic Acid	δ 3.78 (3H, s), 6.45 (1H, d), 6.78 (1H, dd), 7.25 (1H, dd), 7.62–7.79 (2H, 7.68 (dd), 7.72 (d)).	δ, 6.47 (d, H-beta), 7.17 (d, H2), 7.67 (d H-alfa), 3.90 (s, OMe5)	δ 7.27 (H2, H6), 3.81 (H-OMe5), 7.48 (H-alfa), 6.36 (H-beta)	NA	NA	δ 3.99 (s, H-OMe5), 6.56 (m, H-beta), 7.15 (s, H-beta), 7.42 (d, H-alfa)
p-coumaric acid	δ 6.45 (1H, d), 6.90 (2H, ddd), 7.56 (2H, ddd), 7.74 (1H, d).	δ 7.45 (d, H2, H6), 6.80 (d, H3, H5)	δ 6.79 (H2, H6), 7.52 (H3, H5)	δ 6.94 (d), 7.58 (d)	NA	δ 6.95 (d, H2, H6), 7.42 (d, H2, H5)
Resveratrol	δ 6.17 (1H, dd), 6.75 (2H, dd), 6.93 (2H, ddd), 7.04–7.26 (4H, 7.11 (d), 7.18 (d), 7.20 (ddd)).	δ 7.35 (d, H2′, H6′), 6.45 (d, H2, H6)	NA	δ 6.67 (d), 6.92 (d), 6.96 (s), 7.14 (s)	NA	δ 7.15 (d, H2, H6)
Quercetin	δ 6.27 (1H, d), 6.44 (1H, d), 7.16 (2H, ddd), 7.47 (2H, ddd).	δ 7.74 (d, H2′), 7.64 (dd, H6′),	NA	NA	δ 7.53 (dd, H6′)	7.42 (d, H6′)
Isorhamnetin	δ 3.80 (3H, s), 6.27 (1H, d), 6.44 (1H, d), 6.73 (1H, dd), 7.39 (1H, dd), 7.69 (1H, dd).	NA	NA	NA	δ 6.94 (1H, d, H-5′), 3.84 (3H, s, O-CH3);	δ 3.99 (s, H-OMe), 6.95 (d, H5′)

* [22] H−NMR Prediction, nmrdb.org, Binev, Y., Marques, M.M., Aires-de-Sousa, J. Prediction of 1H NMR coupling constants with associative neural networks trained for chemical shifts. *J. Chem. Inf. Model.*
**2007**, *47*, 2089–2097.

**Table 3 pharmaceuticals-16-01110-t003:** Phenolic compounds quantified by HPLC-DAD from the two types of plant extract.

Polyphenolic Acids (mg/mL)										Flavonols (mg/mL)			Heterosides (mg/mL)		Flavones (mg/mL)
Type of Extract	Gallic Acid	Chlorogenic Acid	Caffeic Acid	Ferrulic Acid	Cinnamic Acid	p-Coumaric Acid	Ellagic Acid	E-Resveratol	Z-Resveratol	Kaempferol	Quercetin	Isorhamnetin	Astragalin	Rutin	Scutellarein
P1	1.2603 ± 0.04	0.0079 ± 0.0004	0.0184 ± 0.0008	0.0457 ± 0.0012	0.0170 ± 0.0005	0.02575 ± 0.0009	0.6350 ± 0.0014	* NA	* NA	0.2264 ± 0.0003	0.0578 ± 0.0014	0.0247 ± 0.0021	1.1978 ± 0.0007	0.1478 ± 0.0029	0.6508 ± 0.0027
P2	0.1679 ± 0.0007	0.0045 ± 0.0004	* NA	0.0140 ± 0.0141	0.0028 ± 0.0029	* NA	0.107025 ± 0.0011	0.0019 ± 0.0002	0.0011 ± 0.0001	0.181 ± 0.001	0.009 ± 0.0003	* NA	0.5414 ± 0.0055	0.1906 ± 0.001	* NA

* NA—data not available due to a low quantification limit.

**Table 4 pharmaceuticals-16-01110-t004:** *Sempervivum ruthenicum* Koch xanthine oxidase inhibitory activity.

	Inhibitory Activity (%)						
Extract	100 μg/mL	80 μg/mL	60 μg/mL	50 μg/mL	40 μg/mL	20 μg/mL	IC_50_ μg/mL
P1	82.22 ± 0.45	72.74 ± 0.18	65.62 ± 0.28	63.63 ± 0.14	60.69 ± 0.68	53.52 ± 0.84	11.58 ± 0.36
P2	92.70 ± 0.22	88.78 ± 0.27	74.50 ± 0.36	71.49 ± 0.26	66.70 ± 0.44	56.62 ± 0.59	5.977 ± 0.35
Allopurinol	99.78 ± 0.12	92.83 ± 0.22	82.07 ± 0.44	76.73 ± 0.22	70.82 ± 0.17	58.57 ± 0.78	0.927 ± 0.32

**Table 5 pharmaceuticals-16-01110-t005:** *Sempervivum ruthenicum* Koch albumin denaturation inhibitory activity.

Extract	Concentration μg/mL					IC_50_
	125	250	500	750	1000	
P1ET50	47.598 ± 0.235	56.910 ± 0.514	69.306 ± 0.942	83.066 ± 0.521	92.260 ± 0.155	138.84 ± 5.722
P2ET50	47.361 ± 0.691	55.279 ± 0.714	66.548 ± 0.623	81.999 ± 0.896	92.052 ± 0.257	164.09 ± 9.916
Acetylsalicylic acid	35.973 ± 0.490	46.323 ± 0.420	59.282 ± 1.201	79.330 ± 1.990	92.734 ± 0.448	330 ± 7.821

**Table 6 pharmaceuticals-16-01110-t006:** HPLC assay results for the formulated oral patches (results are expressed as mg/10 cm^2^ bio-adhesive formulation).

Formulation	Gallic Acid	Caffeic Acid	Ferulic Acid	Astragalin	Kaempferol	Quercetin	Rutin	Vanillin
P1	0.4696 ± 0.0031	0.0041 ± 0.0002	0.0067 ± 0.0002	0.4100 ± 0.0023	0.1484 ± 0.0009	0.0279 ± 0.0011	0.0755 ± 0.0021	NA
P2	0.2497 ± 0.0027	0.0027 ± 0.0003	0.0029 ± 0.0003	0.2209 ± 0.0014	0.1001 ± 0.0012	0.5484 ± 0.0019	NA	0.0219 ± 0.0002

**Table 7 pharmaceuticals-16-01110-t007:** Results of the saliva antioxidant capacity.

% DPPH Inhibition (Average ± SD)						
Smokers				Non-Smokers		
Time (Minutes)	Control	P1	P2	Control	P1	P2
0	7.81 ± 0.11	7.83 ± 0.11	7.94 ± 0.11	4.57 ± 0.12	4.66 ± 0.12	4.35 ± 0.12
15	5.06 ± 0.12	6.48 ± 0.11	7.46 ± 0.11	3.91 ± 0.12	2.56 ± 0.12	2.15 ± 0.12
60	7.68 ± 0.11	12.56 ± 0.11	10.43 ± 0.11	4.40 ± 0.12	8.64 ± 0.11	7.16 ± 0.11
90	7.50 ± 0.1	12.98 ± 0.15	11.60 ± 0.11	4.73 ± 0.13	11.12 ± 0.12	8.71 ± 0.04
120	7.53 ± 0.12	13.68 ± 0.02	12.12 ± 0.07	4.67 ± 0.1	11.98 ± 0.04	10.18 ± 0.08

**Table 8 pharmaceuticals-16-01110-t008:** Dilutions used to draw the calibration curves for the two types of plant extract.

P1		P2	
Concentration (mg/mL)	Absorbance (290 nm)	Concentration (mg/mL)	Absorbance (215 nm)
0.5100	3.4532	0.525	2.7961
0.2550	1.9281	0.35	2.0127
0.1270	0.9290	0.2625	1.5142
0.0510	0.4270	0.175	0.9128
0.0250	0.1493	0.131	0.6975
0.0000	0.0000	0	0

**Table 9 pharmaceuticals-16-01110-t009:** Bio-adhesive patch formulation.

Type of Extract	Methylcellulose (mg)	Gelatin (mg)	Polyvinylpyrrolidone (mg)	Pectin (mg)	Propylene Glycol (mg)	Plant Extract (mg)
P1	260	100	20	20	100	60
P2	260	100	20	20	100	60
Concentration (%)	46.42	17.85	3.57	3.57	17.85	10.71

## Data Availability

The data presented in this study are available upon request from the corresponding author. The data are not publicly available due to Romanian legislation restrictions regarding studies that are presented in PhD theses.

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
