# Peer review of "Semperivium Ruthenicum Koch Extract-Loaded Bio-Adhesive Formulation: A Novel Oral Antioxidant Delivery System for Oxidative Stress Reduction"

_pharmaceuticals, 2023, doi:10.3390/ph16081110_

Round 1

Reviewer 1 Report

The work described and the paper are well structured. However, manuscript requires some little revisions. Below are reported the suggestions for the authors:

1.     In the Abstract, the non-smokers group is reported twice in the results part. Please correct the correspondence between the mean values of saliva antioxidant activity and the groups.

2.     In my opinion, it is better that plant extract paragraph is reported as a Method and not as Materials.

3.      Please, add the number of the paragraph “Plant extracts maximum absorbance”.

4.      In paragraph 2.7, should be added the concentration used for gelatin and methylcellulose.

5.     I recommend to report the extract loading efficiency of our bio-adhesive patches to better understand their potential.

Author Response

Dear reviewer,

Thanks again for your suggestions, which were really helpful in improving the quality of our manuscript.

Regarding the new requirements:

  1. In the Abstract, the non-smokers group is reported twice in the results part. Please correct the correspondence between the mean values of saliva antioxidant activity and the groups.

We corrected:

The non-smoking group had a mean saliva antioxidant activity of 7.86±0.66% while the smoking group had a mean value of 4.53±0.15%.

  1. In my opinion, it is better that plant extract paragraph is reported as a Method and not as Materials.

Replaced

  1. Please, add the number of the paragraph “Plant extracts maximum absorbance”.

Added

  1.  In paragraph 2.7, should be added the concentration used for gelatin and methylcellulose.

We have added a line to Table 2, detailing the concentrations used to formulate the patch

Table 2. Bio-adhesive patch formulation.

Type of extract

Methylcellulose

(mg)

Gelatin

(mg)

Polyvinylpyrrolidone

(mg)

Pectin

(mg)

Propylene glycol

(mg)

Plant extract

(mg)

P1

260

100

20

20

100

60

P2

260

100

20

20

100

60

Concentration (%)

46.42

17.85

3.57

3.57

17.85

10.71

  1. I recommend to report the extract loading efficiency of our bio-adhesive patches to better understand their potential.

We have reported the total polyphenol loading efficiency in section 3.5.3. Uniformity of content.

After conducting a spectrophotometric assay at 290 nm and 215 nm, for the P1 and P2 formulations, respectively, the extract content for the P1 formulation was determined to be 6.9592±0.004 mg, and for the P2 formulation 7.0117±0.003 mg. However, upon assessing the total polyphenols using the Folin-Ciocîlteu method, the total polyphenol content of the formulations was quantified as 1.467±0.002 mg Pir/10 cm2 oral patch for the P2 formulation and 5.2443±0.004 mg Pir/10 cm2 oral patch for the P1 formulation, respectively. Based on these results, and on the calculated total phenol content of the dry plant extracts, the loading efficiency for the total polyphenols was calculated to be 95.38±0.24% for the P1 formulation and 95.75±0.33% for the P2 formulation, respectively.

The HPLC assay revealed the presence of several polyphenols, including gallic acid, caffeic acid, ferulic acid, astragalin, kaempferol, quercetin, and rutin. After integration, a total of 1.124 mg polyphenols were quantified from the P2 formulation, and 1.1422 mg from the P1 formulation, respectively. A representative chromatogram and detailed quantification of the identified polyphenols can be seen in Figure 7 and Table 8.

Table 8. HPLC assay results for the formulated oral patches (results are expressed as mg/10 cm2 bio-adhesive formulation)

Formulation

Gallic acid

Caffeic acid

Ferulic acid

Astragalin

Kaempferol

Quercetin

Rutin

Vanillin

P1

0.4696±0.0031

0.0041±0.0002

0.0067±0.0002

0.4100±0.0023

0.1484±0.0009

0.0279±0.0011

0.0755±0.0021

NA

P2

0.2497±0.0027

0.0027±0.0003

0.0029±0.0003

0.2209±0.0014

0.1001±0.0012

0.5484±0.0019

NA

0.0219±0.0002

Figure 7. Representative HPLC chromatogram for the oral bioadhesive formulations. A. P1; B.P2

Thank you again for your support in improving the quality of the article.

Reviewer 2 Report

The work is very interesting and brings new information regarding the use of extractive products with important content in compounds with an antioxidant action, in the treatment of periodontitis.

However, I request the authors to make some clarifications, as follows:

1. In section 2.8.2., Uniformity of content

The determination was made on the pharmaceutical dosage form; after dispersion in water, the maximum absorbance was assessed at 2 wavelengths, for the 2 types of extracts.

Previously, the author determined the maximum absorbance of the plant extract, using a UV-VIS spectrometry method, drawing calibration curves. How was the interpretation made, taking into account the fact that the component polymers can absorb at appropriate wavelengths (e.g. PVP - Polyvinylpyrrolidone - at 207 nm)? Is it certain that there is no overlap between the absorbance related to the polymers in the formulation and the components in the extract?

Perhaps it was safer to quantify the polyphenol content on the respective dispersions, by the HPLC method used for the extract. What followed this determination on the pharmaceutical dosage form? The content in total polyphenols? Their stability?

2. Also, considering the use of Polyvinylpyrrolidone in the formulation, I consider that the abbreviation "PVP" attributed to one of the extracts is not appropriate.

3.Section 3.3.: Where the text refers to Table 5 and Figure 5.

4.Section 3.5.1. Mass uniformity

4.28±0.148 mg What does it refer to? To the oral patches with a 1 cm2 surface area?

5. Section 3.5.5. In vitro release studies

Please reformulate and explaine more clearly. n>1 means zero-order kinetics.

Author Response

Dear reviewer,

Thanks again for your suggestions, which were really helpful in improving the quality of our manuscript.

Regarding the new requirements:

  1. In section 2.8.2., Uniformity of content

The determination was made on the pharmaceutical dosage form; after dispersion in water, the maximum absorbance was assessed at 2 wavelengths, for the 2 types of extracts.

Previously, the author determined the maximum absorbance of the plant extract, using a UV-VIS spectrometry method, drawing calibration curves. How was the interpretation made, taking into account the fact that the component polymers can absorb at appropriate wavelengths (e.g. PVP - Polyvinylpyrrolidone - at 207 nm)? Is it certain that there is no overlap between the absorbance related to the polymers in the formulation and the components in the extract?

Perhaps it was safer to quantify the polyphenol content on the respective dispersions, by the HPLC method used for the extract. What followed this determination on the pharmaceutical dosage form? The content in total polyphenols? Their stability?

As per your revision, we proceeded to quantify the total polyphenols from the pharmaceutical dosage form using the Folin-Ciocilteu method described in the work. We drew a new calibration curve in the 80ug/mL-5 ug/mL concentration range and used a 10 cm2 piece of formulation to complete the assay. The results were reported. Also, we ran a HPLC analysis after dissolving 560 mg polymer formulation (85.795 cm2 containing 60 mg dry plant extracts) in 100 mL of double distilled water. The identified peaks were reported and the concentration was calculated.

After conducting a spectrophotometric assay at 290 nm and 215 nm, for the P1 and P2 formulations, respectively, the extract content for the P1 formulation was determined to be 6.9592±0.004 mg, and for the P2 formulation 7.0117±0.003 mg. However, upon assessing the total polyphenols using the Folin-Ciocîlteu method, the total polyphenol content of the formulations was quantified as 1.467±0.002 mg Pir/10 cm2 oral patch for the P2 formulation and 5.2443±0.004 mg Pir/10 cm2 oral patch for the P1 formulation, respectively. Based on these results, and on the calculated total phenol content of the dry plant extracts, the loading efficiency for the total polyphenols was calculated to be 95.38±0.24% for the P1 formulation and 95.75±0.33% for the P2 formulation, respectively.

The HPLC assay revealed the presence of several polyphenols, including gallic acid, caffeic acid, ferulic acid, astragalin, kaempferol, quercetin, and rutin. After integration, a total of 1.124 mg polyphenols were quantified from the P2 formulation, and 1.1422 mg from the P1 formulation, respectively. A representative chromatogram and detailed quantification of the identified polyphenols can be seen in Figure 7 and Table 8.

Table 8. HPLC assay results for the formulated oral patches (results are expressed as mg/10 cm2 bio-adhesive formulation)

Formulation

Gallic acid

Caffeic acid

Ferulic acid

Astragalin

Kaempferol

Quercetin

Rutin

Vanillin

P1

0.4696±0.0031

0.0041±0.0002

0.0067±0.0002

0.4100±0.0023

0.1484±0.0009

0.0279±0.0011

0.0755±0.0021

NA

P2

0.2497±0.0027

0.0027±0.0003

0.0029±0.0003

0.2209±0.0014

0.1001±0.0012

0.5484±0.0019

NA

0.0219±0.0002

Figure 7. Representative HPLC chromatogram for the oral bioadhesive formulations. A. P1; B.P2

  1. Also, considering the use of Polyvinylpyrrolidone in the formulation, I consider that the abbreviation "PVP" attributed to one of the extracts is not appropriate.

We changed PVU=P1 and PVP=P2

3.Section 3.3.: Where the text refers to Table 5 and Figure 5.

At the begining of paragraph

The HPLC assay conducted on the plant extract revealed a significant amount of bioactive compounds, including polyphenols, flavonols, heterosides, and flavones, as illustrated in Figure 5 and Table 5.

4.Section 3.5.1. Mass uniformity

4.28±0.148 mg What does it refer to? To the oral patches with a 1 cm2 surface area?

We corrected

After examining the weight parameters, it was found that the patches had a con-sistent weight, with an average of 4.28±0.148 mg/cm2 for the P1 formulation and 4.27±0.116 mg/cm2 for the P2 formulation.

  1. Section 3.5.5. In vitro release studies

Please reformulate and explaine more clearly. n>1 means zero-order kinetics.

We reformulated

The release exponent, n, is used to determine the mechanism of drug release from the polymer matrix. A value of n = 0.5 indicates that the drug release is governed by diffusion, while a value of n > 1 indicates a Super Case II, which indicates an extreme form of transport where the outer layer of gel limits the vitreous nucleus, which prevents the axial swelling of the gel and produces tension of compression on the nucleus. As the polymeric interface moves to the nucleus, the increasing tension will lead to the nucleus breaking. The value of n can be used to predict the drug release mechanism and optimize the drug release rate and duration [37].

In this study, both formulations showed non-Fickian (anomalous) diffusion of the active principles from the polymer matrix, as both values of n were >1 (Figure 8). This suggests that the release mechanism is not solely dependent on the diffusion of the active principles through the polymer matrix, but also on other factors such as swelling, erosion, and relaxation of the polymer matrix, and corresponds to a zero-order kinetics release type. When n>1, the release rate decreases more slowly over time, indicating a slower rate of drug diffusion from the matrix. This suggests that the drug is bound more tightly to the polymer, and the release mechanism is influenced by factors such as chain relaxation or swelling of the polymer.

Thank you again for your support in improving the quality of the article.

Round 2

Reviewer 2 Report

I agree with the changes made.